# Lac Operon Boolean Models: Dynamical Robustness and Alternative Improvements

Marco Montalva-Medel [1,*], Thomas Ledger [1,2], Gonzalo A. Ruz [1,2] and Eric Goles [1]

[1] Facultad de Ingeniería y Ciencias, Universidad Adolfo Ibáñez, Av. Diagonal Las Torres 2700, Peñalolén, Santiago 7941169, Chile; tledger@uai.cl (T.L.); gonzalo.ruz@uai.cl (G.A.R.); eric.chacc@uai.cl (E.G.)
[2] Center of Applied Ecology and Sustainability (CAPES), Santiago 8331150, Chile
* Correspondence: marco.montalva@uai.cl

**Abstract:** In Veliz-Cuba and Stigler 2011, Boolean models were proposed for the *lac* operon in *Escherichia coli* capable of reproducing the operon being OFF, ON and bistable for three (low, medium and high) and two (low and high) parameters, representing the concentration ranges of lactose and glucose, respectively. Of these 6 possible combinations of parameters, 5 produce results that match with the biological experiments of Ozbudak et al., 2004. In the remaining one, the models predict the operon being OFF while biological experiments show a bistable behavior. In this paper, we first explore the robustness of two such models in the sense of how much its attractors change against any deterministic update schedule. We prove mathematically that, in cases where there is no bistability, all the dynamics in both models lack limit cycles while, when bistability appears, one model presents 30% of its dynamics with limit cycles while the other only 23%. Secondly, we propose two alternative improvements consisting of biologically supported modifications; one in which both models match with Ozbudak et al., 2004 in all 6 combinations of parameters and, the other one, where we increase the number of parameters to 9, matching in all these cases with the biological experiments of Ozbudak et al., 2004.

**Keywords:** lac operon; catabolite repression; bistability; Boolean network; dynamic; attractor; steady state; limit cycle; update schedule





## 1. Introduction

The *lac* operon in *Escherichia coli* is a paradigmatic example of a genetic regulation system involving the interaction of positive and negative regulatory molecules. The system encodes a pathway for lactose catabolism that is hierarchically controlled by glucose availability, and it has been reported to exhibit a bistable behavior, since the catabolic genes are either uninduced (OFF) or induced (ON) in a single cell, depending on previous activation history and specific extracellular lactose and/or glucose concentrations [1,2]. The *lac* operon has been used as a model system for gene regulation since its initial description in [3], where the concept of an operon was first introduced. Since the molecular components of the regulatory system have been well characterized, it is an excellent candidate for global analysis and modeling. The operon includes three genes involved in the uptake and catabolism of lactose and/or structurally similar sugars. The first gene, *lacZ*, encodes for the enzyme *β*-galactosidase, which converts lactose to allolactose and to subsequent catabolic intermediates; *lacY*, the second structural gene of the pathway, encodes for lactose permease, a membrane transporter for lactose uptake. Finally, the product of gene *lacA* is an acetyltransferase that takes part in the degradation and excretion of non-lactose sugars that may be misrouted through the lactose degradation pathway.

The fundamental states of the *lac* operon are described schematically in Figure 1. In brief, when a low level of extracellular lactose occurs, a strong binding of the LacI protein to the operator sequences strongly represses lac gene expression, thus allowing only a very low amount of mRNA to be transcribed (Figure 1a), which can be considered as an OFF

state. However, even at this low expression level, a few LacY permease molecules can reach the cytoplasmic membrane and take up any eventual lactose that appears outside the cells. The presence of low amounts of LacZ in the cytoplasm leads to transformation of this lactose into *allolactose*, the true natural inducer of the *lac* operon. When this happens, allolactose interacts allosterically with the LacI repressor protein, decreasing its affinity for operator sequences and releasing the negative control of the *lacZ* promoter, rendering the system into an ON state (Figure 1b). Positive feedback loops, like the one described here for the lac system, have been proposed as a typical property of bistable systems [4].

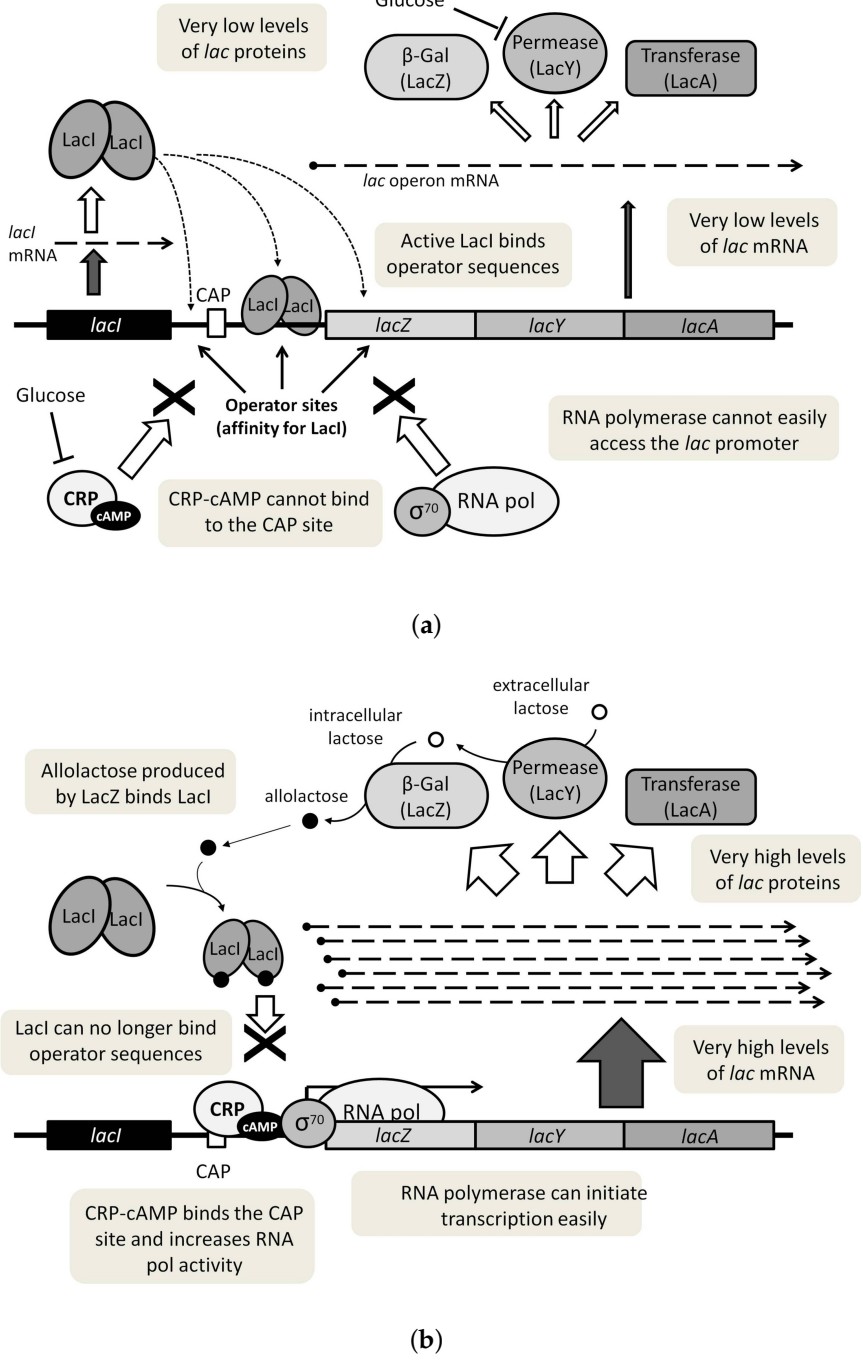

(a)

(b)

**Figure 1.** (**a**) The *lac* operon OFF state is characterized by transcription of low mRNA levels, due to the strong binding of the LacI repressor to operator sequences. (**b**) The lac operon ON state involves transcription of high mRNA levels and increased production of the different lac proteins.

On the other hand, an essential feature of the regulatory system is that high extracellular glucose concentrations can also inhibit the expression of lac genes, a phenomenon traditionally considered as a type of *catabolite repression* that is directly mediated by the glucose molecule. Such a repression has usually been assumed to involve the transcriptional activator protein Crp (cAMP receptor protein), which also takes part in lac gene expression as a cAMP-dependent positive regulator (Figure 1). In the repression model, glucose uptake by the cells produces a drastic reduction in the intracellular cAMP levels, negatively affecting Crp activity and abolishing lac gene expression, which would explain the metabolic preference of glucose over lactose in *E. coli* [5]. An additional explanation of the catabolic hierarchy of glucose is given by the *inducer exclusion* effect, which involves glucose-mediated inhibition of lactose uptake, by means of a direct reduction of LacY permease activity [6]. This would greatly contribute to the regulatory effect of glucose by preventing the inducer molecule to reach the LacI repressor.

Bistability is normally defined as a condition where a system can respond to the same external signal or input in two different manners, depending on the internal state and/or the previous history of the system. However, true bistability requires the system to switch in an all-or-none way between alternative fixed points, without the appearance of intermediate states or a "quantitative" cellular response. Although a quantitative continuous response to the presence of lactose, or the gratuitous inducer thiomethyl-$\beta$-D-galactoside (TMG), was initially shown in experiments measuring the average response of whole culture populations, containing several millions of cells, observations with individual bacteria showed unequivocally that each cell responds in a discrete manner, switching alternatively among the OFF and ON states [1,2], with virtually no evidence of intermediate responses. This shows that bistability can be a characteristic of the *lac* system. Using genetically engineered *E. coli* reporter cells, gratuitous inducers and sophisticated experiments, the authors in [7] showed a discrete hysteretic response of lac gene expression in individual bacteria when pre-incubated in the presence or absence of extracellular lactose and then transferred to fresh medium with different lactose concentrations. Their results showed that, depending on the cell's previous history, the response profile to lactose in the fresh medium varied significantly within a specific concentration range, while outside its limits, cells would respond independently of their previous incubation conditions: Any concentration lower than 3 µM would not lead to *lacZ* induction, whereas any concentration higher than 30 µM would induce the reporter gene. However, most values within the 3–30 µM window were found to induce *lacZ* expression only for lactose preincubated cells. This *window of bistability* was very clearly defined in the absence of extracellular glucose but when this sugar was present, the concentration window moved towards higher lactose concentrations (see Figure 2c in [7]).

Thus, the above all-or-none and bistable behavior that characterizes the lac system makes it suitable for discrete modeling. In particular, Boolean models have been used successfully to analyze genetic regulatory systems on numerous occasions [8–10]. They are useful for the analysis of large regulatory networks when mechanistic details underlying are scarce, insufficiently defined, or even controversial, as it has been discussed that, in such models, the interaction type (inhibition or activation) and network topology are enough for capturing dynamics of gene networks [11,12], without the need of estimating parameters, thereby reducing model complexity. For example, they are being used extensively to represent interactions revealed by high throughput sequencing technologies in the context of systems biology [13,14]. Although some of these assumptions make them less applicable for certain biological processes, they have proved very valuable for predicting the behavior of genes during bacterial metabolism [15] or in the context of interspecies interactions [16]. Even more, other researchers have been able to reproduce bistability for eukaryotic cell differentiation [17], surfactant production in bacteria [18] and microbial signaling [19] using Boolean models.

In this context, the authors of [20] modeled the lac regulation system using a Boolean network that was able to reproduce the bistability of the system under certain but not all

of the conditions described in [7]. Figure 2 displays a scheme of this model, where $G_e$ and $(L_e, L_{em})$ are parameters that account for different concentration levels of glucose and lactose, respectively. This allows to assign specific values as inputs in order to predict the final outcome of the network in terms of fixed points (steady states) that are consistent with the operon being ON, OFF, or in a bistable condition. When the nodes are updated in a parallel regime, the model can reproduce the experimental results in [7] when extracellular glucose is low or absent. However, the predicted outcome of the model differs from the actual behavior of the lac system when lactose and glucose concentrations are high, a feature that limits the usefulness of this initial model for representing lactose fermentation by *E. coli*.

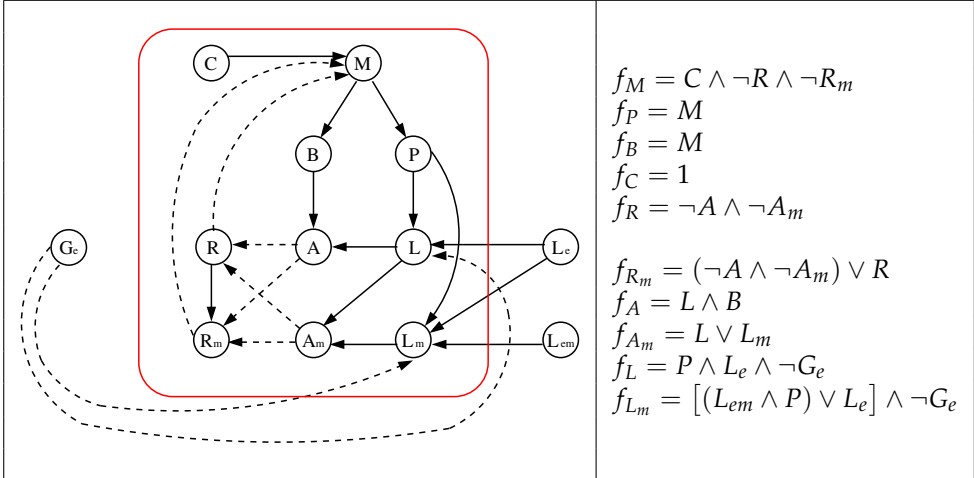

$$f_M = C \wedge \neg R \wedge \neg R_m$$
$$f_P = M$$
$$f_B = M$$
$$f_C = 1$$
$$f_R = \neg A \wedge \neg A_m$$

$$f_{R_m} = (\neg A \wedge \neg A_m) \vee R$$
$$f_A = L \wedge B$$
$$f_{A_m} = L \vee L_m$$
$$f_L = P \wedge L_e \wedge \neg G_e$$
$$f_{L_m} = \big[(L_{em} \wedge P) \vee L_e\big] \wedge \neg G_e$$

**Figure 2.** The original model of [20] (**left**) and its local Boolean functions (**right**). A solid (dashed) edge represents an activation (inhibition). We use the red box to differentiate the parameters of the model (outside) from its variables (inside).

In this work, we first analyze the dynamics of the Boolean models considered but under all different updating schemes in order to know how robust they are in the sense of whether or not new attractors appear. In general, this is a difficult task because the number of different updates schemes associated to a network grow exponentially with its size [21]. Fortunately, there are currently mathematical tools that can significantly reduce this analysis [22]. Secondly, we propose improvements for the performance of the model considered by a rational redesign of the specific Boolean functions in order to predict the operon behavior in each of the sugar combinations tested experimentally in [7], including combined scenarios of low, medium and high concentrations of glucose and lactose. Furthermore, we have taken into account current biological data, regarding catabolic repression and inducer exclusion mechanisms, to bring function definitions up to date.

The paper is organized as follows; Section 2 summarizes the basic mathematical concepts and definitions used along the manuscript, Section 3 contains the most relevant aspects taking into account of [20], in Section 4 our main mathematical results are presented, in Section 5 alternative improvements for the studied models are proposed. Finally, we discuss on our findings in the conclusions section.

## 2. Mathematical Background

A *Boolean Network (BN)* is a couple $(G, F)$, where $G = (V, E)$ is a finite directed graph named *interaction digraph* (e.g., Figure 2(left)), $V$ is a set of $n$ *nodes* ($n$ also called the *network size*), $E \subseteq V \times V$ is the set of *edges* and $F = (f_1, ..., f_n) : \{0,1\}^n \rightarrow \{0,1\}^n$ is a Boolean function composed by $n$ local functions $f_i : \{0,1\}^n \rightarrow \{0,1\}$ such that $x \in \{0,1\}^n \rightarrow x_i = f_i(x) \in \{0,1\}$ ($x$ is called a *configuration* while the value of the variable

$x_k$ associated with node $k$ is known as an *state*). Besides, the local functions $f_i$ depends only on variables $x_j$ such that $(j, i) \in E$ (e.g., Figure 2(right)).

Among the different ways that the states of a BN can be updated is the family of *(deterministic) update schedules* [22] defined as functions $s : V \to \{1, ..., n\}$ such that $s(V) = \{1, ..., m\}$ for some $m \leq n$. In particular, the well-known *parallel* (or *synchronous*) and *sequential* (or *asynchronous*) update schedules are obtained when $m = 1$ (i.e., all states are updated at the same time) and $m = n$ (i.e., all states are updated at different times but following a predetermined order), respectively. We denote by $S_n$ the set of deterministic update schedules that exist for a BN of size $n$. If the states of a configuration $x \in \{0, 1\}^n$ are updated according to a given update schedule $s$, a new configuration $x' = F(x) \in \{0, 1\}^n$ will be obtained. The change from $x$ to $x'$ can be represented by the arc $x \to x'$ and is known as *transition* from $x$ to $x'$. Thus, the $2^n$ configurations and their respective transitions give rise to the *dynamics* of the system which can be represented in a *state transition graph* Furthermore, because it is finite, the dynamics has limit behaviors called *attractors*, which are of 2 types:

- *Fixed point (or steady state)*: configuration $x \in \{0, 1\}^n$ such that $F(x) = x$.
- *Limit cycle*: sequence of configurations $x^0, ..., x^{l-1} \in \{0, 1\}^n$ pairwise distinct such that $F(x^j) = x^{j+1}$, for all $j = 0, ..., l - 2$, $F(x^{l-1}) = x^0$ and $l > 1$ is a integer named the *length* of the limit cycle.

The set of configurations that converge to a specific attractor is called the *attraction basin*.

## 3. The *lac* Operon Boolean Models to Be Considered: Main Aspects and Choice Justification

In [20] the authors proposed different Boolean models explaining the bistability of the *lac* operon in *Escherichia coli*. These are BNs that have the following characteristics in common:

1. All of them have qualitative behaviors that match very well with the experiments performed in [7].
2. The interaction digraph is composed by nodes that represent mRNA, proteins, and sugars. Its edges represent the type of interaction between the nodes (activation/inhibition).
3. The dynamic is obtained considering the parallel update schedule.

Next, we will specify which are the models of [20] that we choose. Hence, we will summarize the main points that their authors studied and with which we will work from Section 4 onwards. Finally, we justify our choices.

### 3.1. The Chosen Models of Veliz-Cuba and Stigler 2011: Those without Catabolite Repression

We will consider the alternative model without catabolite repression proposed in [20] which differs from the initial one presented in that work only in the definition of the local function $f_C$; in the initial one $f_C = \neg G_e$ while in the alternative model without catabolite repression $f_C = 1$ (see our justification in Section 3.4). In addition, we will also consider its reduced version (without catabolite repression) and, from now on, we will refer to this alternative model without catabolite repression and its reduced version as the *original* and *reduced* one, respectively. We show the details of these models in Figures 2 and 3, where we point out that the networks have been drawn "disarming" those of [20] which consider some pairs of nodes as a single node, but the reader can easily check that our networks and the corresponding of [20] are equivalents.

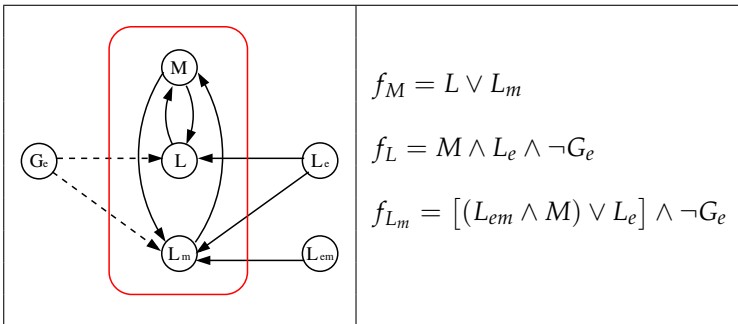

**Figure 3.** The reduced model of [20] (**left**) and its local Boolean functions (**right**). Dashed/solid edges represent inhibitions/activations.

### 3.2. The Dynamics Produced by the Original and Reduced Models

According to [20], the original model (Figure 2) assumes two concentration of glucose, low ($G_e = 0$) or high ($G_e = 1$), and three concentrations of extracellular lactose, low, medium or high, represented by the pairs of parameters $(L_e, L_{em}) = (0,0), (0,1)$ and $(1,1)$, respectively (the pair $(L_e, L_{em}) = (1,0)$ has no meaning, therefore is not considered). This gives a total of six different combinations of parameters and, consequently, each one will have its corresponding dynamic. Here, a configuration corresponds to the vector $(M, P, B, C, R, R_m, A, A_m, L, L_m) \in \{0,1\}^{10}$ and, for purposes of relating the dynamics to the results of [7] (see details in Section 3.3), only its steady states matter, i.e., limit cycles (if they exist) are ignored. Besides, when $(M, P, B) = (1,1,1)$ (when $(M, P, B) = (0,0,0)$) it will be said that the operon is ON (OFF). Abusing this notation, henceforth, we will call ON (OFF) to a steady state that verifies the above condition.

Therefore, if the dynamics for each of the six combinations of parameters are obtained and only are observed its steady states, then, there exist the following four cases:

**Case 1:** $G_e = 1$. Any configuration eventually converges to the unique fixed point OFF $= (0,0,0,1,1,1,0,0,0,0)$.

**Case 2:** $G_e = L_e = L_{em} = 0$. Any configuration eventually converges to the unique fixed point OFF$= (0,0,0,1,1,1,0,0,0,0)$.

**Case 3:** $G_e = 0 \wedge L_e = L_{em} = 1$. Any configuration eventually converges to the unique fixed point ON$= (1,1,1,1,0,0,1,1,1,1)$.

**Case 4:** $G_e = L_e = 0 \wedge L_{em} = 1$. Any configuration eventually converges to one of the two fixed points; OFF$= (0,0,0,1,1,1,0,0,0,0)$ or ON$= (1,1,1,1,0,0,0,1,0,1)$, i.e., bistability is obtained.

Notice that case 1 implicitly covers three of the six combinations of parameters (those obtained when high concentration of glucose is combined with low, medium and high concentration of extracellular lactose, respectively), all three having the same dynamic. The remaining three combinations of parameters are those obtained in cases 2, 3 and 4, each with a different dynamic.

We summarize the four cases in Table 1 and their respective dynamics in Figure 4.

Regarding the reduced model (Figure 3), four cases similar to the original one are obtained, which can be summarized using the same Table 1. It should be mentioned that, in this reduced model, a configuration is of the form $(M, L, L_m) \in \{0,1\}^3$, and the operon is ON (OFF) when $M = 1$ ($M = 0$). In the Figure 5, we show the dynamics of its four cases.

Notice that although only 3 different dynamics are generated (because its cases 1 and 2, involving 4 out of 6 combinations of parameters, generate a single dynamic), they are still in accordance with the summary of Table 1.

**Table 1.** The possible cases in the original model for each of the six combinations of parameters.

| Glucose ⟍ Lactose | Low $G_e = 0$ | High $G_e = 1$ |
|---|---|---|
| Low: $(L_e, L_{em}) = (0,0)$ | Case 2 (OFF) | Case 1 (OFF) |
| Medium: $(L_e, L_{em}) = (0,1)$ | Case 4 (Bistability) | Case 1 (OFF) |
| High: $(L_e, L_{em}) = (1,1)$ | Case 3 (ON) | Case 1 (OFF) |

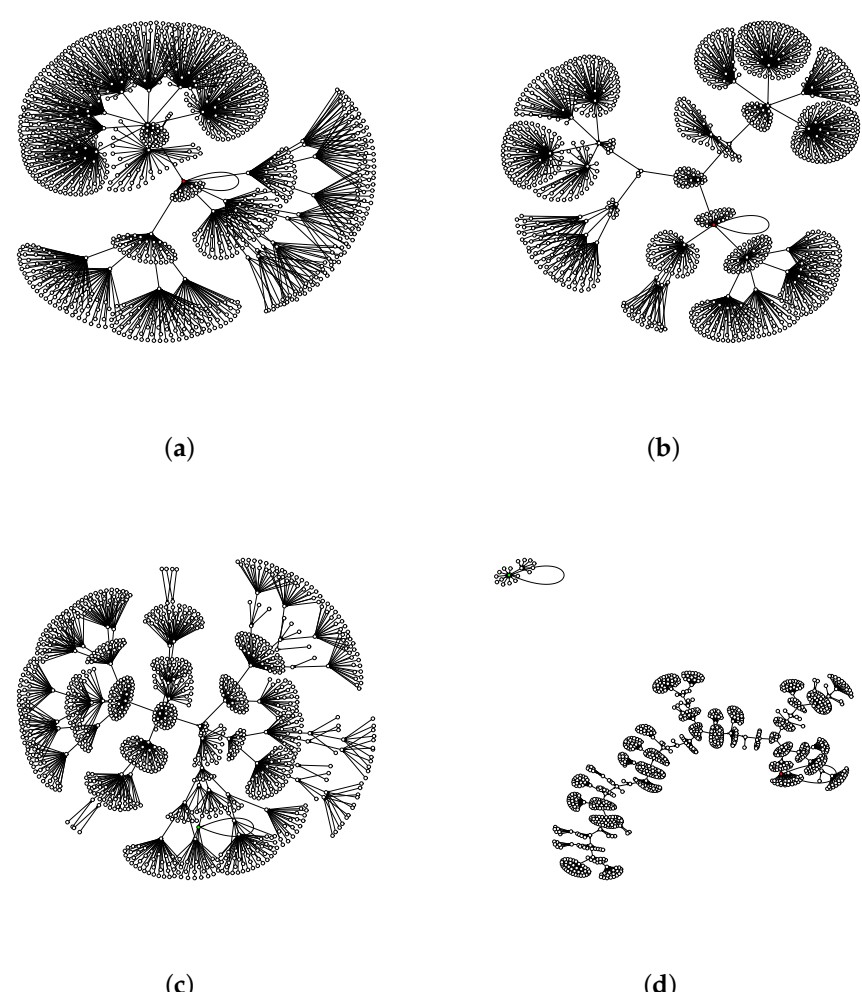

(a)  (b)

(c)  (d)

**Figure 4.** State transition graphs of the original model under parallel update schedule for: (**a**) Case 1, (**b**) Case 2, (**c**) Case 3 and (**d**) Case 4 (bistability appears). The circles represent all the $2^{10} = 1024$ configurations $(M, P, B, C, R, R_m, A, A_m, L, L_m) \in \{0,1\}^{10}$, in particular, the red one correspond to the steady state OFF while the green one correspond to the steady state ON. The sizes of its attraction basins are 1024 for cases 1, 2 and 3 and 1006 and 18 for the OFF and ON steady states of case 4, respectively.

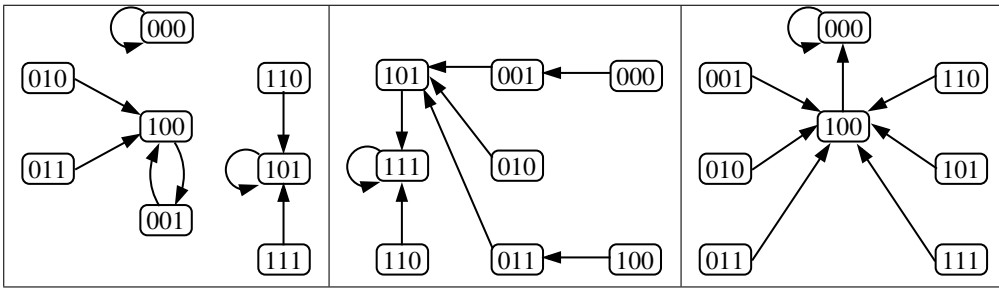

**Figure 5.** State transition graphs of the reduced model under parallel update schedule. Case 4 (**left**): low glucose and medium lactose. Case 3 (**center**): low glucose and high lactose. Cases 1 and 2 (**right**): the other four combination of parameters.

### 3.3. Stochastic Simulations in the Original Model Which Largely Coincide with the Biological Experiments of Ozbudak et al., 2004

The bistability of the *lac* Operon, according to [23], means that it must be a region of bistability, i.e., for a range of parameters, a set of "cells" may have both, OFF and ON, fixed points. In [20], this is accomplished by considering stochasticity in the uptake of the inducer in the following way, considering that glucose is absent ($G_e = 0$) while that extracellular lactose is given as the discretization of a random variable $\mathcal{L} \sim N(\mu, \sigma)$ taken from a normal distribution with $\mu$ equal to the mean of $\{0.6, ..., 2.8\}$ incremented by 0.2 and $\sigma = 0.3$ (these values are chosen in order to be consistent with those measured in the biological experiments of [7]). Then, the extracellular lactose ($L_e, L_{em}$) is given as a function of $\mathcal{L}$ defined by

$$(L_e, L_{em}) = \begin{cases} (0,0) = \text{low}, & \text{if } \mathcal{L} \leq 1 \\ (0,1) = \text{medium}, & \text{if } 1 \leq \mathcal{L} < 2 \\ (1,1) = \text{high}, & \text{if } 2 \leq \mathcal{L} \end{cases}$$

Below, we summarize the stochastic experiment carried out in [20]:

(1) Starting with the normal distribution $N(0.6, 0.3)$ for $\mathcal{L}$, to generate randomly a set of values for $\mathcal{L}$ and calculate for each of them the corresponding value for the pair ($L_e, L_{em}$).

(2) Assuming $G_e = 0$, to find which are the steady states of the dynamic obtained for each value ($L_e, L_{em}$) of (1), it is simply one of the three possibilities showed in the second column of Table 1; OFF, bistable (i.e., OFF and ON) or ON.

(3) To repeat (1) and (2) but for $N(\mu, 0.3)$ with $\mu \in \{0.8, 1.0, 1.2, ..., 2.8\}$.

The results of this experiment were those that matched with the biological experiments of [7] in 5 out of 6 combinations of parameters described in Section 3.2. In the remaining one, where glucose and lactose concentrations are high (see Table 1), the original model (as well as the reduced one) predicts the operon being OFF while the biological experiments of [7] show a bistable behavior.

### 3.4. Our Justification for the Choice of Models without Catabolite Repression

One of the most relevant outcomes of the Boolean models proposed in [20] is that the presence of glucose at a medium or high concentration would be able to completely repress the expression of the *lac* mRNA, which is a direct consequence of the inhibition function $f_C = \neg G_e$, which aims to take into account one of the proposed aspects of catabolite repression. However, this assumption fails to consider the well-described fact that glucose is not able to shut down LacZ expression in bacteria when the operon has been pre-induced experimentally [24]. Furthermore, the whole concept of glucose repression mediated by a drop in cAMP and subsequent lack of the CRP-cAMP activator complex has been under critical revision [25–28] based on the following compelling challenges: experimental data indicate that intracellular cAMP concentrations are similar when lactose, glucose, or a mixture of both substrates is present at concentrations lower than 300 μM and that cAMP abundance is only reduced by a factor of 5–8 when external glucose concentrations

are above 300 μM [29,30], which is higher than the bistability window explored in [7]. Exogenous addition of cAMP did not abolish glucose-mediated inhibition of lac gene expression, even though the metabolite was shown to enter the cells [31]. On the other hand, when the repressor protein LacI is inactivated by deletion of the *lacI* gene, no catabolite repression of LacZ is observed, indicating that this process relies solely on the activity of the repressor protein, shutting down transcriptional activity by means of inducer exclusion [32]. Finally, the CRP-cAMP activator complex is also required for the expression of the PtsG glucose uptake transporter [33], which would imply that, if this pathway of catabolite repression were valid, glucose would eventually inhibit its own transport, which has never been observed. This inconsistency with experimental data and the relevant objections discussed above justify our choice of the models without catabolite repression of [20], where $f_C = 1$, to account for the current view of glucose-induced *lac* repression.

## 4. Results: Dynamical Robustness of the Original and Reduced Models

Our goal is to study the dynamics of the original and reduced models not only for the parallel update schedule but for the whole family of deterministic update schedules defined in Section 2. As evidenced in [21], this set grows exponentially with the network size. In particular, its sizes are $|S_3| = 13$ and $|S_{10}| = 102{,}247{,}563$ for the reduced and original models, respectively.

Observe that it is easy to deduce that the steady states OFF and ON obtained in the dynamics of the original, and reduced models will also continue to appear with any other deterministic update schedule because their are known to be invariant under update schemes; however, limit cycles could also appear making the desired modeling less robust. We will analyze this aspect below.

### 4.1. Dynamical Robustness of the Original Model

We begin by giving a short known Lemma that allows us to prove that the dynamics of the first three cases mentioned in Section 3.2 are highly robust for any deterministic update schedule.

**Lemma 1.** *Let $N = (G, F)$ be a Boolean network updated by the update schedule s. If G is acyclic and it has no loop [i.e., an edge of the form (i,i)], then the attractors are only fixed points.*

**Proof.** It is known that the nodes of an acyclic digraph can be ordered in $m$ layers, being the first and last those that have the root and leaves nodes, respectively, such that the arcs belongs between the $i$-layer and the $j$-layer, for $1 \leq i \leq m - 1$ and $i < j$. Since there are no loops, the dynamic evolves fixing the states of the nodes, in the worst case, from the first to the last layer. Therefore, the only possible attractor is a fixed point.  □

4.1.1. Cases 1, 2 and 3 for the Original Model

**Proposition 1.** *The dynamics associated with the cases 1, 2 and 3 of the original model of Section 3.2 have no limit cycle, whatever the update schedule considered.*

**Proof.** Let $s$ be an arbitrary update schedule and $(M, P, B, C, R, R_m, A, A_m, L, L_m) \in \{0,1\}^{10}$ an initial configuration at time $t = 0$. We will show that in every case, we can obtain one of the acyclic digraphs without loops showed in Figure 6:

**Cases 1 and 2.** It is easy to check that $C = 1$ and $L = L_m = 0, \forall t \geq 1$. This implies that, $A = A_m = 0, \forall t \geq 2$. Therefore, we have the left digraph of Figure 6.

**Case 3.** Notice that $C = L_m = 1, \forall t \geq 1$. Therefore, we have the following sequence of implications; $A_m = 1, \forall t \geq 2 \Rightarrow R = 0, \forall t \geq 3 \Rightarrow R_m = 0, \forall t \geq 4$. Therefore, we have the right digraph of Figure 6.

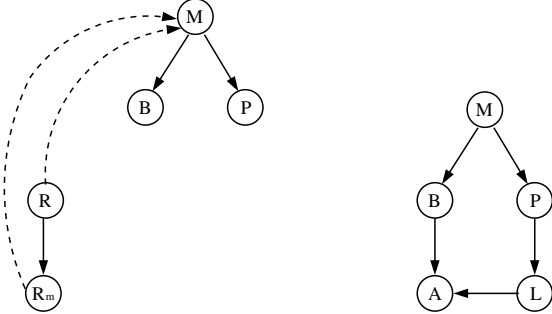

**Figure 6.** Acyclic digraphs without loops obtained when considering the parameters of the cases 1 (**left**), 2 (left also) and 3 (**right**) of Section 3.2.

Thus, in every case, Lemma 1 assures that the attractors are only fixed points, i.e., there is no limit cycle. □

Proposition 1 implies that for the first 3 cases, and whatever be the update schedule, the attraction basin of its dynamics will be the whole space of configurations $\{0,1\}^{10}$, as summarized in Table 2.

**Table 2.** Size of the attraction basin associated with the dynamics of cases 1, 2 and 3 for the original model (Section 3.2), and considering any of the $|S_{10}| = 102,247,563$ possible update schedules.

| Case | Attractor | OFF | ON |
|---|---|---|---|
| 1: $G_e = 1$ | | 1024 | 0 |
| 2: $G_e = L_e = L_{em} = 0$ | | 1024 | 0 |
| 3: $G_e = 0 \wedge L_e = L_{em} = 1$ | | 0 | 1024 |

### 4.1.2. Case 4 for the Original Model

It is easy to see that for an arbitrary update schedule and a initial configuration $(M, P, B, C, R, R_m, A, A_m, L, L_m) \in \{0,1\}^{10}$ at $t = 0$, we will have that $C = 1$ and $L = 0$, $\forall t \geq 1$. This implies that $A = 0$, $\forall t \geq 2$, and we obtain the digraph of Figure 7 that has three cycles; one of length 6 and two of length 5. Therefore, we cannot apply the previous Lemma.

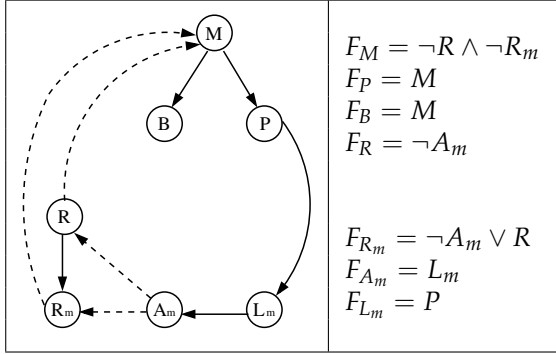

$$F_M = \neg R \wedge \neg R_m$$
$$F_P = M$$
$$F_B = M$$
$$F_R = \neg A_m$$

$$F_{R_m} = \neg A_m \vee R$$
$$F_{A_m} = L_m$$
$$F_{L_m} = P$$

**Figure 7.** Digraph obtained when considering the parameters of case 4 (**left**) of Section 3.2 and its local functions after considering $G_e = L_e = L = A = 0$ and $L_{em} = C = 1$ (**right**).

We will analyze its dynamical behavior through a detailed study of all its different dynamics by using the algorithms developed in [22] that, roughly speaking, list all the sets of schemes that generate exactly the same dynamics, significantly reducing the number of dynamics to analyze. These algorithms have been shown to be effective in obtaining new and relevant information for the study of concrete biological networks [34,35] as well as in the field of the cellular automata theory [36–41].

Simply, we generate one dynamic for each of the above representative update schedules and summarize its results in Table 3.

**Table 3.** Average size of the attraction basin for case 4 of the original model (Section 3.2) and calculated over the set $S_{10}$, $FP \subseteq S$ (set of update schedules whose dynamic have only steady states) and $LC \subseteq S$ (set of update schedules whose dynamic have limit cycles). Its respective sizes are $|S_{10}| = 102,247,563$ (100%), $|FP| = 71,891,966$ (70.3%) and $|LC| = 30,355,597$ (29.7%).

| Case 4: $G_e = L_e = 0 \wedge L_{em} = 1$ (Bistability) | | | |
|---|---|---|---|
| **Attractors** | *S* | *FP* | *LC* |
| OFF | 684.7 | 908.9 | 153.7 |
| ON | 124.4 | 115.1 | 146.5 |
| Limit cycles | 214.9 | 0 | 723.8 |
| Total | 1024 | 1024 | 1024 |

We can observe that less than 30% of all the dynamics present limit cycles and with a balanced proportion between ON/OFF basins, but they are smaller than those associated to the limit cycles, which are on average 5 times larger than that of ON and OFF, respectively. On the other hand, more than 70% of all the dynamics did not have any limit cycle, i.e., they only had the fixed points OFF and ON (being, on average, about 8 times greater the OFF basin than the ON basin).

In Figure 8 we exhibit one example of the dynamical behavior that may have the original model when the update schedule considered is slightly different from the parallel one. There are, in addition to the fixed points ON an OFF, three limit cycles of length 4 and one of length 2.

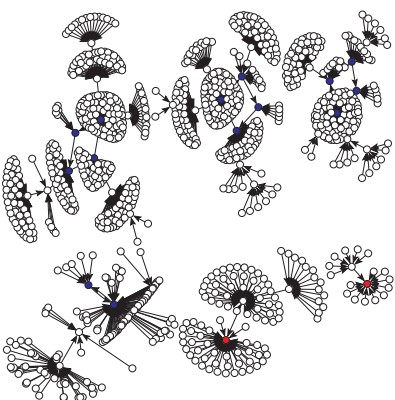

**Figure 8.** State transition graph of the original model under the parameters of case 4 and considering that the first nodes $M, P, B, C, R, A, L, L_m$ are updated (in parallel) and then the nodes $R_m, A_m$ (also in parallel). The size of the attraction basins are 18 and 98 for the steady states ON and OFF, respectively (both in red), and 908 considering the four limit cycles (in blue).

*4.2. Dynamical Robustness of the Reduced Model*

Clearly, the dynamical robustness of the reduced model is quick and straightforward to analyze, so we only summarize its results.

4.2.1. Cases 1, 2 and 3 for the Reduced Model

**Proposition 2.** *The dynamics associated with the cases 1, 2 and 3 of the reduced model of Section 3.2 have no limit cycle, whatever the update schedule considered.*

The dynamics for the above three cases also are quite robust for any deterministic update schedule, and this is summarized in Table 4.

**Table 4.** Size of the attraction basin associated with the dynamics of cases 1, 2 and 3 for the reduced model (Section 3.2) and considering any of the $|S_3| = 13$ possible updates schedules.

| Attractor | OFF | ON |
| --- | --- | --- |
| Case | | |
| 1: $G_e = 1$ | 8 | 0 |
| 2: $G_e = L_e = L_{em} = 0$ | 8 | 0 |
| 3: $G_e = 0 \wedge L_e = L_{em} = 1$ | 0 | 8 |

### 4.2.2. Case 4 for the Reduced Model

Now the exhaustive analysis boils down to just 13 possible update schedules and the results are summarized in Table 5.

**Table 5.** Average size of the attraction basin for case 4 of the reduced model (Section 3.2) and calculated over the set $S_3$, $FP$ and $LC$ (see definitions in Table 3). Its respective sizes are $|S_3| = 13$ (100%), $|FP| = 10$ (77%) and $|LC| = 3$ (23%).

| Case 4: $G_e = L_e = 0 \wedge L_{em} = 1$ (Bistability) | | | |
| --- | --- | --- | --- |
| **Attractors** | **S** | **FP** | **LC** |
| OFF | 2.5 | 2.8 | 1.3 |
| ON | 4.6 | 5.2 | 2.7 |
| Limit cycles | 0.9 | 0 | 4 |

Observe that about 77% of the dynamics have only the fixed points OFF and ON, but, in this case the average size of the ON attractor basin is almost 2 times bigger than that of OFF. The same occurs in the other 23%. Furthermore, the OFF and ON basins add up, on average, exactly the same as that of the limit cycles.

Examples of the dynamical behavior of the reduced model exhibiting limit cycles are those of Figure 5(left).

## 5. Alternative Improvements for the Studied Models

As mentioned at the end of Section 3.3, of the 6 parameter combinations, $(G_e, L_e, L_{em}) = (1, 1, 1)$ –meaning glucose and lactose concentrations in high levels (see Table 1)–it is the only one that does not matches with the experimental data (see Figure 2c in [7]) where bistability should be observed again, instead of the operon being OFF as obtained from the models. Therefore, the first improvement naturally consists in solving that in both models. The second one will consist of increasing the parameters of the original model from 6 to 9, matching in all these cases with the biological experiments of [7].

*5.1. Improvement 1: The Original and Reduced Models Match in All 6 Parameter Combinations with Ozbudak et al., 2004*

Observe that according to what was discussed at the beginning of Section 4, some local functions of both models must necessarily be modified to make the steady states OFF and ON appear at the same time (bistability). On the other hand, by considering carefully the arguments we gave in Section 3.4, the local function $f_C = 1$ is not sufficient to adjust the Boolean model to represent the actual behavior of the system when both glucose and lactose are present in the extracellular environment of *E. coli*. To further refine the response of the Boolean network to the balance of each different sugar and to highlight the relevance of the previous history of the cell, a new definition has been proposed for $f_L$ and $f_{L_m}$, considering the inhibitory effect of glucose on lactose transport will be significant in the short term as long as a low concentration of lactose was initially present, while the opposite will be true if a high lactose concentration was already inside the cell. These changes are shown in Figures 9 and 10.

| | |
|---|---|
| $f_M = C \wedge \neg R \wedge \neg R_m$ | $f_M = C \wedge \neg R \wedge \neg R_m$ |
| $f_P = M$ | $f_P = M$ |
| $f_B = M$ | $f_B = M$ |
| $f_C = 1$ | $f_C = 1$ |
| $f_R = \neg A \wedge \neg A_m$ | $f_R = \neg A \wedge \neg A_m$ |
| $f_{R_m} = (\neg A \wedge \neg A_m) \vee R$ | $f_{R_m} = (\neg A \wedge \neg A_m) \vee R$ |
| $f_A = L \wedge B$ | $f_A = L \wedge B$ |
| $f_{A_m} = L \vee L_m$ | $f_{A_m} = L \vee L_m$ |
| $f_L = P \wedge L_e \wedge \neg G_e$ | $f_L = P \wedge \big[ (L_{em} \wedge \neg G_e) \vee L_e \big]$ |
| $f_{L_m} = \big[ (L_{em} \wedge P) \vee L_e \big] \wedge \neg G_e$ | $f_{L_m} = \big[ (L_{em} \wedge P) \vee L_e \big] \wedge \big[ \neg G_e \vee L \big]$ |

**Figure 9.** The original model (**left**) and the improved original model (**right**), where changes are marked in red.

| | |
|---|---|
| $f_M = L \vee L_m$ | $f_M = L \vee L_m$ |
| $f_L = M \wedge L_e \wedge \neg G_e$ | $f_L = M \wedge \big[ (L_{em} \wedge \neg G_e) \vee L_e \big]$ |
| $f_{L_m} = \big[ (L_{em} \wedge M) \vee L_e \big] \wedge \neg G_e$ | $f_{L_m} = \big[ (L_{em} \wedge M) \vee L_e \big] \wedge \big[ \neg G_e \vee L \big]$ |

**Figure 10.** The reduced model (**left**) and the improved reduced model (**right**), where changes are marked in red.

Besides, we show in Figures 11 and 12 the interaction digraph and the dynamics of the improved original and reduced models, respectively.

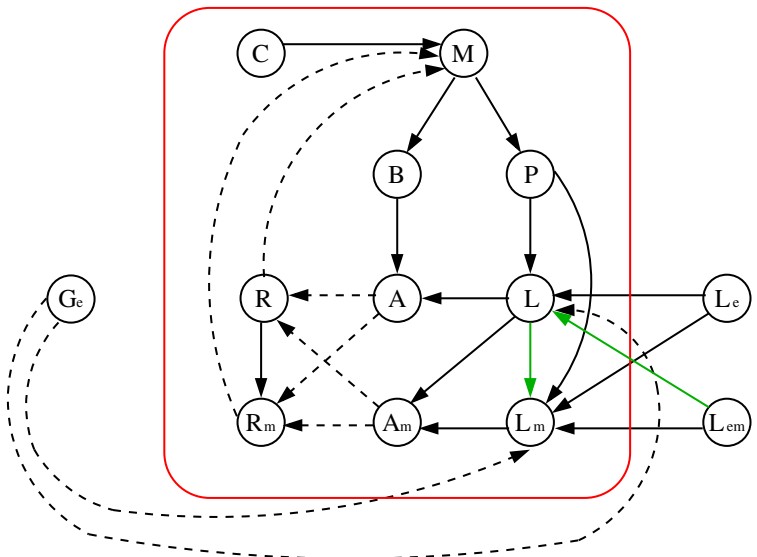

**Figure 11.** Interaction digraph associated to the improved original model.

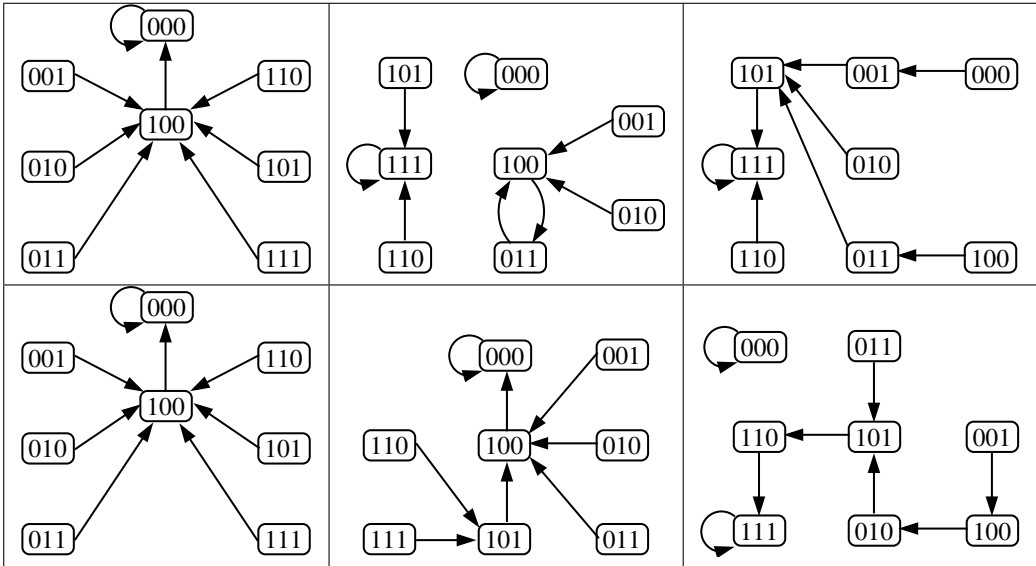

**Figure 12.** Dynamic of the improved reduced model associated with; low glucose and low lactose (**top left**), low glucose and medium lactose (**top center**), low glucose and high lactose (**top right**), high glucose and low lactose (**bottom left**), high glucose and medium lactose (**bottom center**) and high glucose and high lactose (**bottom right**).

Thus, the improved original and reduced models predicts:

1. **Bistability:** when $(G_e, L_e, L_{em}) \in \{(0,0,1),(1,1,1)\}$.
2. **OFF:** when $(G_e, L_e, L_{em}) \in \{(0,0,0),(1,0,0),(1,0,1)\}$.
3. **ON:** when $(G_e, L_e, L_{em}) = (0,1,1)$.

The above is summarized in Table 6, which match for all the 6 combinations of parameters with the biological experiments of [7].

**Table 6.** The possible cases in the improved original and reduced models for each of the six combinations of parameters.

| Glucose Lactose | Low $G_e = 0$ | High $G_e = 1$ |
|---|---|---|
| Low: $(L_e, L_{em}) = (0,0)$ | OFF | OFF |
| Medium: $(L_e, L_{em}) = (0,1)$ | Bistability | OFF |
| High: $(L_e, L_{em}) = (1,1)$ | ON | Bistability |

### 5.2. Improvement 2: (Improved) Original Model Extended to 9 Parameters

As explained in the introduction, there is a *window of bistability* (see Figure 2c in [7]) which justifies that three parameters can also be considered for the glucose (low, medium and high). Taking into account again the arguments given in Section 3.4 as well as those discussed for the first improvement, the following changes are proposed over the improved original model (see Figures 13 and 14).

At this point, we repeat the stochastic experiment described in Section 3.3, but for different glucose values. Its results are shown in Figure 15 and summarized in Table 7.

Observe how the results of this extended model match the entire window of bistability (see Figure 2c in [7]) in each of its 9 combinations of parameters.

$$f_M = C \wedge \neg R \wedge \neg R_m$$
$$f_P = M$$
$$f_B = M$$
$$f_C = 1$$
$$f_R = \neg A \wedge \neg A_m$$
$$f_{R_m} = (\neg A \wedge \neg A_m) \vee R$$
$$f_A = L \wedge B$$
$$f_{A_m} = L \vee L_m$$
$$\color{red} f_L = \Big( P \wedge \big[ (L_{em} \wedge \neg G_e) \vee L_e \big] \Big) \vee \Big( P \wedge \big[ (L_{em} \wedge \neg G_{em}) \vee L_e \big] \Big)$$
$$\color{red} f_{L_m} = \Big( \big[ (L_{em} \wedge P) \vee L_e \big] \wedge \big[ \neg G_e \vee L \big] \Big) \vee \Big( \big[ (L_{em} \wedge P) \vee L_e \big] \wedge \big[ \neg G_{em} \vee L \big] \Big)$$

**Figure 13.** The improved model shown in Figure 9 (right) but with the additional parameter $G_{em}$. Modified local functions appear in red.

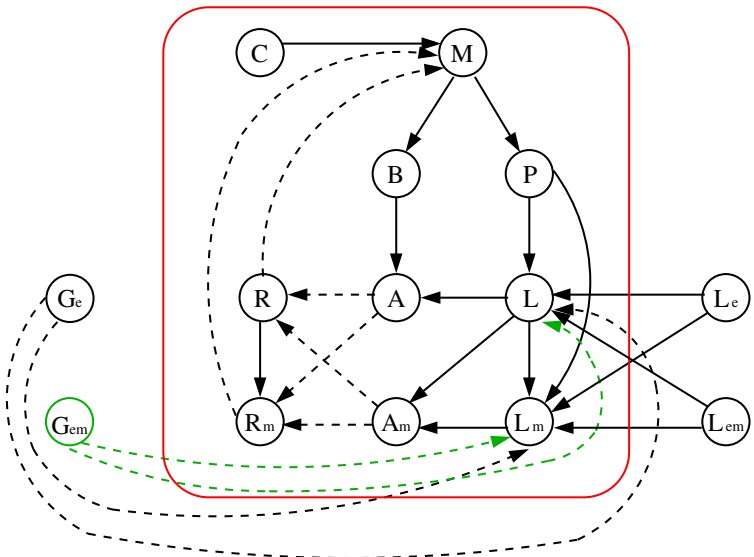

**Figure 14.** Interaction digraph of the improved original model with additional parameter $G_{em}$ and the local functions of Figure 13.

**Table 7.** The possible cases in the 4-parameter model of Figures 13 and 14 for each of the nine combinations of parameters.

| Lactose \ Glucose | Low $(G_e, G_{em}) = (0, 0)$ | Medium $(G_e, G_{em}) = (0, 1)$ | High $(G_e, G_{em}) = (1, 1)$ |
|---|---|---|---|
| Low $(L_e, L_{em}) = (0, 0)$ | OFF | OFF | OFF |
| Medium $(L_e, L_{em}) = (0, 0)$ | Bistability | Bistability | OFF |
| High $(L_e, L_{em}) = (0, 0)$ | ON | ON | Bistability |

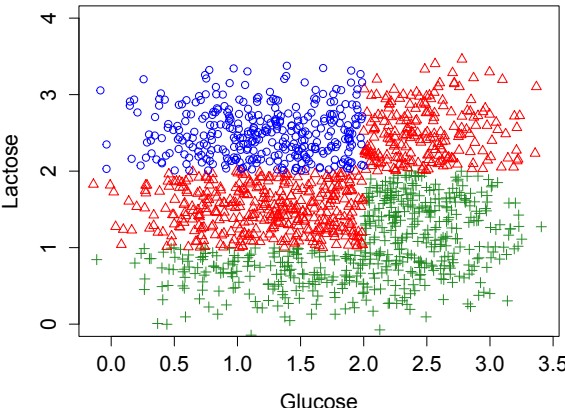

**Figure 15.** Bifurcation diagram for the stochastic experiment performed over the improved original model of Figures 13 and 14. Blue, red and green marks represent dynamics with steady states ON, bistability (OFF and ON at the same time) and OFF, respectively.

## 6. Conclusions

The *lac* Operon Boolean model without catabolite repression and its respective reduced version proposed in [20] were analyzed. These models have the particularity of being simple but capable of reproducing the operon being OFF, ON and bistable for different levels of lactose and glucose, matching very well with biological experiments such as those published in [7]. Unlike the models of [20], the Boolean networks proposed here predict bistability even at high glucose concentrations. This feature has been observed experimentally when inducer (a non-metabolizable lactose analogue) concentrations are also high [7]. Furthermore, our models take into account intermediate glucose concentrations, thus increasing sensibility to changes within the "window of bistability" of the lac system (see more details in the second part below).

In a first part of this paper, we have studied the dynamical robustness of these models where we made the following contributions:

- For the first 3 cases described in Section 3.2 and that included 5 of the 6 combinations of parameters allowed in the original and reduced models, we establish two Propositions proving the non-existence of any limit cycle, whatever the update schedule used.
- For the case 4, where bistability appears, we made an exhaustive analysis of all its possible dynamics generated with any update schedule. Here we detail for both models the average sizes of their attraction basins, the number of dynamics without limit cycles (i.e., only with fixed points) and the number of dynamics with limit cycles (being less than 30% in both models).
- Again in the case 4, its predominant attractor (those that have the bigger attraction basin), changes dramatically; OFF attraction basin being, in average, 8 times bigger than that of ON for the original model while that in the reduced one, the ON basin is almost 2 times bigger than that of OFF.

The above findings allow us to conclude that the effect of the glucose and lactose parameters in the interaction digraph associated to each network breaks its cycles for cases 1, 2 and 3, transformed into an acyclic network that will have only fixed points as attractors and, consequently, being completely robust in these situations. However, in the case of bistability, the role of the update schedule can change this property significantly because limit cycles with large attraction basins can appear. Moreover, this work supports the hypothesis of [20] that network topology is a key factor for qualitative dynamical properties but not for quantitative ones.

As a second part, we have proposed two alternative improvements for the Boolean models studied, with biological support and that involve small modifications in their local functions. In the first improvement, the prediction is corrected for the case in which the parameters of the models represent high glucose and lactose levels, achieving the bistability

observed in the biological experiments of [7]. In this way, with such an improvement, the original and reduced models match perfectly with the above experiments of [7] for all the 6 combinations of parameters. The second improvement increases the possible combinations of parameters for the original model, going from 6 to 9, enriching the dynamics of the models and matching the bistability window observed in [7] with each of the 9 possible combinations of parameters. By keeping in mind that in our models, inducer exclusion can effectively explain operon regulation, which takes into account the current challenges against the glucose-mediated repression model, our results can also be compared with some continuous models such as those of [42], based on differential equations, where bistability windows are displayed in a wide range of glucose concentrations. It is worth mentioning that the main softwares used in this manuscript were Matlab (for the exhaustive analysis of all the different dynamics of each model and which allowed us to build Tables 3 and 5) and RStudio (for the visualization of most of the state transition graphs presented in this work as well as for the stochastic experiment; Figure 15).

Although the lac operon is a classical model and many of its key molecular players were identified a long time ago, our understanding of how these players interact with each other has evolved continuously through the years [43]. Full comprehension of the system requires robust and thorough knowledge of key regulatory features, like catabolite repression, where the view of a direct inhibitory effect of glucose on the cAMP-CRP regulatory system has been challenged during the last decades [25–30]. Furthermore, lac operon bistability is a feature that is hard to reproduce by models in general, and the applicability of a Boolean network including glucose-mediated regulatory systems has only been tried previously in [20]. Our present work shows how the translation of updated mechanistic information about the actual role of glucose into the network allows taking better advantage of a Boolean model to test and reproduce bistability in a way that is faithfully representative of experimental data. This has multiple implications as, in general, bistability has been considered a ubiquitous feature in bacteria, involving several processes [44], and understanding the dynamics of global gene regulatory systems revealed by high throughput technologies has become increasingly complex, thus demanding simpler, robust modelling algorithms, such as Boolean networks. Future models could include additional molecular features of the operon for the lac system, like repressor oligomerization, inducer (lactose) degradation by LacZ activity, and consideration of additional binding sites within the lac promoter region to provide a more detailed description of biological outcomes. Another natural future extension consists of giving a more quantitative approach to the models here studied, similar to what has been done in works such as [45–47].

**Author Contributions:** Conceptualization, M.M.-M. and T.L.; methodology, M.M.-M. and T.L.; software, M.M.-M. and G.A.R.; validation, M.M.-M., T.L., G.A.R. and E.G.; formal analysis, M.M.-M. and T.L.; investigation, M.M.-M., T.L., G.A.R. and E.G.; writing—original draft preparation, M.M.-M., T.L., G.A.R. and E.G.; writing—review and editing, M.M.-M., T.L., G.A.R. and E.G.; visualization, M.M.-M., T.L. and G.A.R.; funding acquisition, M.M.-M., G.A.R. and E.G. All authors have read and agreed to the published version of the manuscript.

**Funding:** This research was partially funded by STIC-AmSud CoDANet project 19-STIC-03 (M.M.-M., E.G.), ANID FONDECYT grant number 1200006 (E.G.), ANID FONDECYT grant number 1180706 (G.A.R.), and ANID PIA/BASAL FB0002 (T.L.).

**Institutional Review Board Statement:** Not applicable.

**Informed Consent Statement:** Not applicable.

**Data Availability Statement:** The data used to support the findings of this study are included within the article.

**Conflicts of Interest:** The authors declare no conflict of interest.

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
