# Peer review of "Lac Operon Boolean Models: Dynamical Robustness and Alternative Improvements"

_mathematics, doi:10.3390/math9060600_

Round 1

Reviewer 1 Report

Montalva-Medel and colleagues presented their work on improving boolean models of lac operon to better correlate with the experimental measurements.  The work can be of interest to some readers, however, the reviewer finds some aspects that need to be improved.

First, the boolean model typically has of limited use in biological systems, and the authors showed a minor improvement over a previous work (ref 1). The authors also mentioned that topological constraints can only give qualitative description.  The authors need to present their case better to clearly show that boolean models have merits than other modeling efforts.

Second, the authors keep attacking catabolite repression but mentions only one or two works that back up their claims. They need to do more thorough literature survey to substantiate their claims. 

Third, lac operon is a classic model that has been worked out many times. The authors need to apply their model to more recent system (even if experimental data are not as substantial), to show that their modeling approach can be general.

In conclusion, the authors need to present their work more clearly and show other examples to showcase that the boolean description is worth considering as models for biological systems. 

Other minor points:

Page 2: histeretical -> hysteretic

Page 6: Case 4: ON and OFF labels are swapped

Page 10: the same dynamics that s

Page 16: over de improved -> over the improved

Page 16: analized -> analyzed 

Reviewer 2 Report

This is a very interesting paper in Lac operon Boolean models. The paper is clearly written and the subject is very well presented.

The reviewer believes that the paper can be accepted for publication. It would be nice to include some possible future extensions in the conclusion.

Some minor grammatical errors can be corrected during the preparation of the camera ready version. Some of these minor errors are the following:

Revise the phrase in line 257: “We will study it dynamical behavior…”

Revise the phrase in line 267: “This latest having…”

In line 327 write “analyzed” instead of “analized”

In line 354 write: “with such an improvement”

Reviewer 3 Report

The authors deal with an interesting approach to the problem of modeling the lac operon of E.coli.

They propose an improvement of a previously published model, using a very interesting approach based on Boolean operators.

The modelization is convincing and sounds very well.

However, the comparison of results with the biological experiment (reported in #[2] of bibliography) is just reported in a small sentence at rows 327-329.

In my opinion, it is not sufficient and I suggest extending the discussion of results in the light of experimental biological evidence. 

Reviewer 4 Report

The paper presents the model based on a Boolean network describing lac operon in Escherichia coli. The paper is well written and is scientifically sound. The mathematical tools used are adequate. It includes all necessary mathematical definitions. My comments:

  1. The most important my comment is related to the models based on differential equations (see, e.g. work of Santillan, 2008 https://www.ncbi.nlm.nih.gov/pmc/articles/PMC2257910/). Are your results comparable with the results with the help of dynamical system with the help of three differential equations? What are your advantages? In my mind, all results of the Santillan's work should be applicable. Can be presented bifurcation diagrams with the help of your model?
  2. Since most of the results are obtained numerically, in my mind, it should be presented the topics of errors of experiments. For example, how many iterations were used to get a limit cycle?
  3. What can you say about the period of the limit cycle? Did you observe two- or higher- period limit cycles?
  4. Some words about the software used are needed.

Round 2

Reviewer 1 Report

The authors addressed most of the previous concerns, and thus, I would recommend publication of the manuscript.